# How health literacy relates to venous leg ulcer healing: A scoping review

**Ayoub Bouguettaya**[1]ORCID*, **Georgina Gethin**[2], **Sebastian Probst**[3,4]ORCID, **Jane Sixsmith**[5], **Victoria Team**[6], **Carolina Weller**[6]ORCID*

1 School of Psychology, University of Birmingham, Birmingham, United Kingdom, 2 School of Nursing and Midwifery, National University of Ireland, Galway, Ireland, 3 School of Health Sciences, HES-SO University of Applied Sciences and Arts Western Switzerland, Geneva, Switzerland, 4 Care Directorate, University Hospital Geneva, Geneva, Switzerland, 5 School of Health Sciences, National University of Ireland, Galway, Ireland, 6 Wound Research Group, School of Nursing and Midwifery, Monash University Melbourne, Victoria, Australia

☯ These authors contributed equally to this work.
* carolina.weller@monash.edu (CW); A.Bouguettaya@bham.ac.uk (AB)

**Data Availability Statement:** These data are the result of a review and therefore all research used for the conclusions are available to all.

**Funding:** The corresponding author (CW) was supported by the grant from the Australian NHMRC

## Abstract

### Background

The level of personal health literacy of patients with venous leg ulcers is likely to affect their ability to self-manage their condition impacting on their adherence to treatment and influences healing and recovery outcomes.

### Objectives

To scope existing research that examined the level of health literacy in venous leg ulcer patients, to identify how this may link to self-management behaviours (particularly physical activity and compression adherence), and venous leg ulcer healing outcomes.

### Methods

This scoping review was based on the PRISMA-ScR six-stage framework. We searched MEDLINE, EMBASE, the Cochrane Library, PsycInfo and Health, Open Grey, and Google Scholar for publications examining general and specific health literacy in those with venous leg ulcers and for those examining any potential links of health literacy with self-management/healing generally, published between 2000–2020. This search was guided by a published protocol; studies that described other types of ulcers or did not examine health literacy were excluded. After applying inclusion and exclusion criteria the initial search identified 660 articles.

### Results

We included five articles. Four studies used randomised controlled trials or experimental designs to test the effect of specific health literacy interventions on venous leg ulcer knowledge, compression therapy use, or healing outcomes. One study was a cross-sectional survey with qualitative elements, assessing health literacy in venous leg ulcer patients.

(NUMBER APP1069329). https://www.nhmrc.gov.au/ he funders had no role in study design, data collection and analysis, decision to publish, or preparation of the manuscript.

**Competing interests:** The authors have declared that no competing interests exist.

Broadly, the research suggested that health literacy was suboptimal amongst those with venous leg ulcers, and health literacy interventions had limited effects on improving key venous leg ulcer specific outcomes.

## Conclusion

This review provides a synthesis of extant literature examining health literacy in patients with venous leg ulcers. We identified a dearth of literature investigating the value of general and specific health literacy interventions in this space. Most importantly, no recent research on general health literacy and venous leg ulcers was identified, despite strong theoretical utility to do so. The few studies identified largely indicated that targeting health literacy of patients with venous leg ulcers is a viable area of research and intervention, encouraging future researchers and clinicians to consider patient health literacy in venous leg ulcer management.

## Introduction

Venous Leg Ulcers (VLUs) are a significant health problem worldwide. VLUs are a chronic skin condition of the lower limbs, often affecting the elderly and those with chronic health illnesses [1, 2]. VLUs affect approximately 1–3% of adults globally [2–4]. Furthermore, these wounds tend to be severe; recent evidence suggests that almost 20% of wounds present in aged community samples are VLUs [5]. The prevalence of VLU increases with age, doubling among those aged over 65 years [4, 6] The typical VLU patient tends to be female, over 65 years old, has a high risk of cardiovascular events, and has obesity [7]. VLU patients often face ongoing health and quality of life problems [6, 8] partially because the treatment itself requires substantial self-management, leading to significant intrapersonal and interpersonal burdens [3, 6, 9].

For decades, the best practice recommendation for those with an active VLU has been consistent-the use of compression therapy and physical activity [10, 11] Research suggests that the use of below-knee multi-component compression is efficacious [12–14] Compression therapy promotes VLU healing by reducing the hydrostatic pressure in lower limbs, enhancing venous return [10] and preventing venous stasis [15] Consistent compression therapy is recommended to prevent VLU recurrence [12–14] While other lifestyle changes are also commonly recommended, such as adequate nutrition, leg elevation and particularly physical activity [10, 16, 17] the most consistent evidence base has been the recommendation to patients to consistently use and self-apply compression garments.

In recent years, there has been renewed interest in the idea of patient health literacy (HL) in how patients adhere to treatments like compression therapy and physical activity. Patient HL is defined as "the ability to obtain, process, and understand basic health information". There is some overlap between HL and health knowledge with knowledge of health, health care and systems identified as an element of HL in some definitions. HL provides a toolkit beyond health knowledge alone that can enable patients to apply their knowledge for successful self-management. It is likely that there is a relationship between patients' knowledge of when and how to use information and the knowledge that they hold. HL is not monolithic; in fact, it is better to see health literacy as health literacies [18] as literacy is related both to the content and the context in which a patient must act. Therefore, here we detail a theoretical distinction between general HL often assessed through population-level surveys [19] and specific HL which deals with health skills and knowledge specific to the condition or disease [20].

"General" HL, as described here, relates to a conceptualisation of HL as a bundle of higher level skills required in order to acquire, understand, and apply health knowledge [18] This is further split in most studies along three key domains: functional literacy (encompassing skills in reading and writing to function), communicative/interactive literacy (extracting/applying new information to changes in circumstances) and critical literacy (advanced skills to critique and use information to control situations and life events when needed; [21]. General HL scales are often used to assess a patient's general capabilities in accessing and navigating health services (e.g., where do you go for medical advice), often for the purpose of directing health policy at a population scale [18, 21]. However, because health literacy often requires more specific skills that do not neatly fit into functional, communicative, or critical domains, focusing on *a* specific health literacy that is activated by specific illness contexts may add to our understanding of the impact of HL on health conditions. Therefore, specific HL scales assess individual capabilities in dealing with a specific condition by examining specific skills and confidence that require integration of communicative, functional, and critical literacy in a specific way. For example, specific HL scales exist for diabetes by extending the HL scales to be more specific toward diabetes management skills, such as capacity to understand insulin management [20, 22].

Both general and VLU-specific HL may affect VLU outcomes through affecting patient's ability to adopt health behaviours. Improvements in general and specific HL may improve patient knowledge and understanding of the benefits of adhering to VLU self-management recommendations [23–25], and support patients to adopt healthy behaviours in line with an agreed management plan. For example, when choosing compression hosiery, patients may rely on HL to critique the options based on their analysis of comfort [26] compared with achieving maximum therapeutic benefit [27]. Furthermore, HL may enhance their compression application skills due to improved understanding of the manufacturer's instructions [14, 28]. Finally, improved HL may influence patient understanding that lifelong compression hosiery is recommended to prevent VLU recurrence [29].

Recent research has reported that people vulnerable to VLUs, those with VLUs, and those with other comorbidities tend to have concurrent unmet HL needs [30]. For example, although the 2015 European HL survey showed that respondents received an average score of 33.8/50 (demonstrating "sufficient" HL), the majority (58%+) of people aged over 66 years had limited HL, compared to less than half of the general population [19]. A possible reason is internet usage. Though internet usage is proportional to increased HL [31], current research consistently reports that older adults prefer to learn from their health care professionals [29], as opposed to independent learning through the use of the internet [32]. Qualitative research has shown that VLU patients often discuss the volume of information and skills that are needed for self-management upon VLU development as a significant burden, as VLU self-management can be complex [23] and demanding [24]. The education needs of VLU patients are not well understood [25] contributing to unmet HL needs. In general, limited HL in adults is associated with reduced adherence to treatment and health recommendations, poorer health outcomes and increased cost of medical treatment [33], especially amongst older adults [34]. Furthermore, checking on patient understanding is not a routine practice for health care professionals [5], although this was recommended in at least one set of international guidelines on VLU care [11].

## Rationale

Limited qualitative studies published in the past indicate that HL may affect VLU patients' self-management capabilities [35–37], yet there have been no recent reviews published

examining the level of HL of patients with VLUs, and the effect it has on patient's adoption of health behaviours. These studies suggest that inadequate HL reduces the likelihood of engaging in VLU compression, but there is also the possibility that those with lower HL may not increase their physical activity in response to a VLU (despite also being in the recommendations). One educational intervention study (N = 20) reported specific health knowledge in VLU is suboptimal, although demonstrated that there is utility in improving health knowledge (and potentially skills) in VLU patients, as they also felt more confident in their VLU management afterwards [38]. However, this study did not examine physical activity rates, and was hampered by a small sample size. Other research [39] on illnesses that share psychological antecedents and risk factors with VLUs (e.g., diabetes; [40, 41]) suggests that understanding HL in these patients can translate into improvements in health.

## Research objectives and present study

The aim of this scoping review was to scope the research examining the level of HL in VLU patients, how this level may link to self-management behaviours/knowledge (particularly physical activity and compression adherence), and VLU healing generally. Given that HL in VLU patients appears less extensively researched than other factors in VLU healing, we opted to conduct a scoping review. This review type was a flexible method for identifying and discussing information useful for answering our research questions, and allowing a holistic presentation of the available literature on this topic [42]. Our primary question was:

1) What levels of HL (both general and specific) have been reported in adults with active or past VLUs across outpatient, home care, community, and inpatient care?

   Our secondary research questions were as follows:

2) Is there any relationship (correlational or experimental) between HL and VLU patient adherence to compression therapy, health knowledge, and physical activity?

3) Is there any relationship between HL and VLU patient healing outcomes?

## Methods

The approach for this scoping review was adapted from the PRISMA [43] guidelines for scoping reviews. We conducted the review in accordance with the PRISMA Extended for Scoping Reviews (PRISMA-ScR) [14] (Table 1). Methods were developed based on guidelines developed by Levac et al. [43] using the six framework stages in the published protocol [44].

Help in formulating and executing this search was provided by the host university's Library services at [redacted] University, [city, location]. This search strategy was registered in the published protocol [44]. No ethics approval was required as we used data from publicly available platforms with no risk of identification of participants.

Using COVIDENCE and EndNote, one author (AB) ran multiple searches through academic search engines (EBSCOhost, PubMed, Open Grey) and public search engines (e.g., Google Scholar) to search through the targeted databases of MEDLINE, EMBASE, the Cochrane Library, PsycInfo and Health, Open Grey, and Google Scholar. This search strategy sought to find papers to answer the three research questions. The search was limited to January 2000 to August 2020, with the most recent search conducted February 2021. The literature search was developed using a combination of medical search headings and free text words. The keywords and search string relevant to Medline via Ovid can be found in S1 Appendix. The full search was conducted using Boolean operators and proximity operators, including wildcards, AND, OR, parentheses, quotations, and more as per the database used. All papers

**Table 1. Included studies.**

| Author, year | Title | Country | Setting | Intervention type | Study population | Aims of the study | Methods | Outcome measures | Results |
|---|---|---|---|---|---|---|---|---|---|
| 1. Protz et al, 2019 | Education in people with venous leg ulcers based on a brochure about compression therapy: A quasi-randomised controlled trial | Germany & Austria | Ambulatory care and medical practices, hospitals and specialised clinical wound practice | Intervention group provided with brochure: Compression therapy–easy and well-fitting; asked to read this at home and complete a knowledge questionnaire at the next visit. Control group completed the questionnaire at their first clinic visit without the brochure. | 136 Patients with VLU > 18 years, German speaking. (IG 68 / CG68) | This study will provide data on how much patients with VLU and related compression therapy benefit from using the described brochure on patient education. It focuses on the improvement of theoretical knowledge, practical abilities, and related skills-subjectively and factually. | RCT | 1. Estimation of subjective knowledge [4-point scale 1 = very good, 4 = inadequate]. 2. Knowledge of effects of compression therapy and risk for venous disease [two questions, five items]. 3. Self-care and vein support activities [two questions, five and six items respectively]. 4. Skin care [two questions]. 5. Decongestion phase [one question]. 6. Care of compression materials [one question]. 7. Donning devices [one question]. | 1. In total, the subjective knowledge in the case group was similar or only a bit higher than in the control group. The six questions addressed the function of the veins, the causes of venous diseases, the effects of compression therapy, the products of compression therapy, the cleansing of compression materials, and the awareness about what the patient can contribute to therapy and prevention. Only in the latter case, the difference between the two groups was significant (p = 0.023) 2. Patients in the case group had significantly more knowledge than patients in the control group in most items except underweight (p .359). 3. In most items, the patients in the case group had significantly more knowledge than the patients in the control group. According to the knowledge about vein sport activities, both groups mainly answered correctly. Nevertheless, for each item, a (much) higher percentage of the control group chose the wrong option. 4. The difference between the two groups was significant in any item (P = .000) with those in the intervention group responding correctly more than the control group. 5. There was no statistical difference between groups in five out of six items with the question son use of multicomponent systems correctly answered more often by the intervention group. 6. The comparison of the case and control groups showed significant differences (P = .000) in the knowledge about donning devices for the easier usage of medical compression stockings. |

*(Continued)*

**Table 1.** (Continued)

| Author, year | Title | Country | Setting | Intervention type | Study population | Aims of the study | Methods | Outcome measures | Results |
|---|---|---|---|---|---|---|---|---|---|
| 2. Gonzalez 2014 | Education project to improve venous stasis self-management knowledge | USA | Out patients specializing in wound care | Education intervention of 45 mins duration; brochure and handout and overview of self-care activities | 30 patients with VLU | To evaluate patients' knowledge of chronic venous disease, venous ulcers, and self-care at baseline, immediately following, 2, and 9 weeks after an educational intervention. | Single group before and after design | 1. Scores on the Checklist for Patient Learning measured at baseline, 2 week and 9week follow-up 2. Wound healing as reported by participants at 2 and 9 weeks. 3. Wound recurrence as reported by participants at 9 weeks. | 1. Statistically significant differences were found when baseline mean scores were compared to scores immediately following intervention, and scores measured at 2 and 9 weeks. 2. Ninety-three percent of wounds were assessed as healing at 2 weeks, and 80% were assessed as healing at 9 weeks 3. The reported venous ulcer recurrence rate at 9 weeks was 50% |
| 3. Clarke-Moloney et al, 2005 | Information leaflets for venous leg ulcer patients: are they effective? | Ireland | Vascular surgery department | Verbal information plus an information leaflet | 20 patients with newly diagnosed VLU | To determine the usefulness of providing written information for patients with leg ulcers. Emphasis on testing the information provided rather than testing the patients. | RCT | 1. Differences in patient's knowledge of their condition and its treatment after 4–6 weeks. | 1. At follow-up there was an overall significant improvement in the amount of correct answers for a number of questions, although there was no significant improvement between the groups. The difference in total scores at baseline and follow-up between the groups was not significantly different (p = 0.83) |
| 4. Gonzalez, 2017 | The effect of a patient education intervention on knowledge and venous ulcer recurrence: results of a prospective intervention and retrospective analysis | USA, Florida | Home Care | A patient education programme. A 45-minute programme delivered in a one-on-one setting in the patients' home | Three groups: A (n = 28) Follow up assessment of education programme, B (n = 22) newly enrolled group C (n = 45) retrospective chart review representing a control group | To understand the long-term efficacy of patient education and to assess a cause-effect relationship between education and the prevention of venous ulcer recurrence. | Three parallel groups, non-randomised | 1. Knowledge scores assessed using Checklist for patient learning [13-point scale with 13 indicating best outcome]. 2. Healing rates 3. Recurrence rates | 1. Unpaired t-tests comparing groups A and B at post-intervention (P = .121), 2 weeks (P = .544), 9 weeks (P = .433), and 36 weeks following the intervention (P = 0.687) indicate no substantial difference in the scores between the groups. Learning retention for group A at 36 weeks following the educational intervention showed it was significant for the entire Checklist for Patient Learning (P = .041) as well as for the subscales *disease knowledge* (P = .039) and *self-care knowledge* (P = .022). In group B increases in scores were noted for the entire Checklist for Patient Learning and for the disease and self-care knowledge subscales: 36-weeks disease subscale (P = .020) and self-care knowledge subscale (P = .021). 2. Wound healing at 9 weeks was Group A 80%; Group B 86% and not reported for group C 3. Recurrence at 9 and 36 weeks respectively for group A was 50% and 50%; Group B 36% and 55% and Group C 69% and 31%. |
| 5. Finlayson et al 2010 | The impact of psychosocial factors on adherence to compression therapy to prevent recurrence of venous leg ulcers | Australia | Patient clinics or a hospital department and a community-based clinic | No intervention. This was a follow up survey of those previously diagnosed with a VLU that healed between 12–36 months prior to the study | 122 participants | 1. To identify participants knowledge regarding their condition and the self-care activities they undertook to prevent recurrence 2. To determine which demographic or psychosocial factors influenced adherence to compression therapy | Cross-sectional survey and retrospective chart review | 1. Knowledge of cause of their condition 2. Self-care activities 3. Demographic health and psychosocial factors influencing adherence to compression hosiery | 1. Full range of knowledge and self-care activities reported. No comparison groups and no baseline or follow-up scores, so only scores for one time point reported. |

went through a quality assessment. This assessment of the included studies was independently conducted by AB and VT, and any differences in the quality assessment were resolved through a consensus meeting and discussion We used the Critical Appraisal Skills Program (CASP) Checklist for RCTs (see [45]) for the included RCTs by Clarke Moloney et al [46] and Protz et al [47] the National Institutes of Health (NIH) Quality Assessment Tool of controlled intervention studies [48] for the included non-randomised controlled intervention by Gonzalez [49]; the NIH Quality Assessment Tool for observational cohort and cross-sectional studies see [48] for a cross-sectional study by Finlayson et al; and NIH Quality Assessment Tool for before-after (pre-post) studies with no control group for the pre- and post- educational intervention study by Gonzalez [50]. All CASP assessments can be found in S1 File.

Authors also had to make a judgement on whether or not HL was specific or general, as most papers did not clarify this distinction; all authors worked together to reach a consensus on this categorisation.

## Results

### Search results

The literature search retrieved 660 citations; 29 duplicates were excluded resulting in 629 titles (Fig 1). Overall, 23 were considered eligible for potential inclusion. Following full text assessment, we further excluded 18 studies, leaving five remaining. Fig 1 describes this process, and Table 1 details the findings. Due to the low number and study heterogeneity, no synthesis was attempted. The inconsistent way in which HL was described by authors led to some overlap between HL and prior health knowledge. Some studies detailed skills as part of health knowledge, and therefore these studies were retained. Therefore, this assessment of quality of evidence was left to a discussion between authors instead, and represented through a table (Table 1).

### Summary of studies

**Levels of HL in VLU patients.** Of the five studies, one examined specific HL in VLU patients [51]. This was a cross-sectional (survey) study with qualitative elements [51]. The cross-sectional study examined specific HL in VLU patients (N = 122), in relation to their knowledge of the cause of their condition, their self-care activities, and demographic healthy/psychosocial factors relating to their adherence to compression hosiery. The majority of participants (58%) either did not know the cause of their VLUs or cited incorrect causes, suggesting specific HL in this cohort was low. Most participants did not wear their compression stockings daily (53%), with 20% not wearing their compression at all. A similar proportion replaced their compression hosiery less than twice a year. Multiple regression analysis further showed that the level of knowledge was strongly associated with wearing compression hosiery ($\beta$ = 1.92, SE = .61, p = .002, partial $\eta^2$ = .10).

**HL and VLU compression, exercise, and VLU knowledge.** Four studies examined the effect of HL interventions on VLU compression/healing outcomes or VLU knowledge [47–50]. Of these, all used specific HL interventions designed to educate patients about VLUs, and how to self-manage. None used general HL interventions designed to improve general patient knowledge and efficacy. Of these four, none examined the effect of the HL intervention against compression, and two examined the effect of HL intervention against healing. None measured the effect of HL interventions against physical activity in VLU patients, or assessed a link between HL and physical activity. All four examined the change in health knowledge compared to baseline.

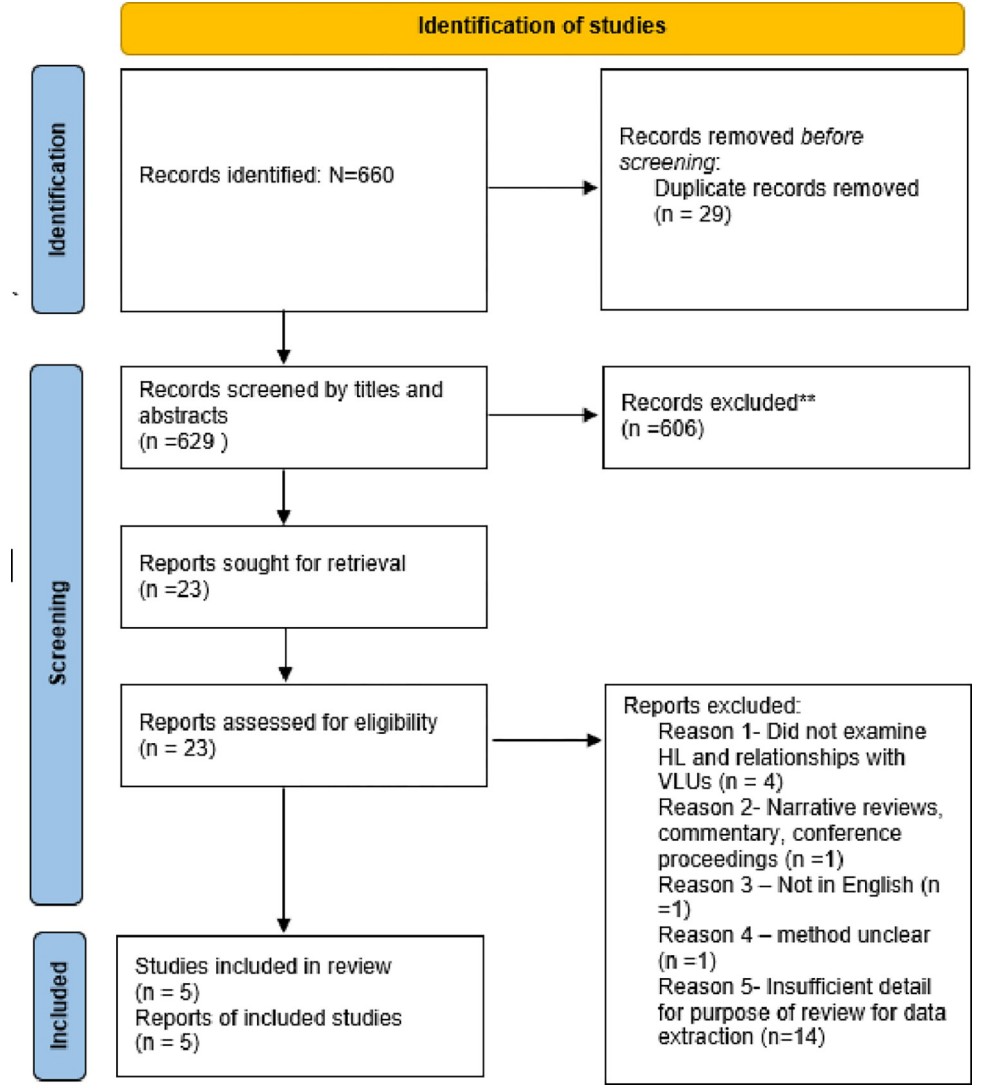

**Fig 1. PRISMA flow diagram.**

HL interventions designed to improve specific HL against VLUs for patients were broadly successful. A quasi-randomised controlled trial (N = 136) in German and Austrian VLU patients demonstrated this by providing half of patients with a brochure on skills, and testing them on their VLU specific HL later [47]. This intervention showed that this type of intervention increased HL about self-care, donning devices, skincare, and knowledge of the effects of compression in the experimental group compared to the control group, but did not appear to improve HL in subjective knowledge estimation, multicomponent systems, and various questions on veins, causes of diseases, and cleaning of materials.

Another randomised controlled trial (N = 20) examined the effect of adding a leaflet to patient information for newly diagnosed VLU patients in Ireland, after a verbal explanation of their condition [46]. They assessed specific HL in relation to VLUs as their main outcome, finding that overall, there was a significant improvement in specific HL at follow-up for both the control and leaflet group. However, there was no significant difference between the

patients receiving the leaflet versus no leaflet at both follow-ups, suggesting the leaflet was ineffective.

**HL and patient healing.** Two American studies by the same author provided an intervention to improve VLU patient knowledge and examined VLU wound healing as an outcome. The first [49] tested a patient education program on VLUs for VLU patients (N = 30). The authors used a custom tool to assess VLU specific HL on self-care and disease process (asking 13 questions, with a maximum score of 13 if all answered correctly). The authors also assessed VLU healing/recurrence. Results showed there were significant, sustained gains in specific HL knowledge at multiple timepoints compared to pre-educational programs, and recurrence was lower than in other studies, the effect of the educational intervention on wound healing was not directly assessed. The second study [50] used the same tools, methods, and outcomes (N = 95), but used a non-randomised design to compare three groups. A retrospective chart review was used to act as a control group (n = 45), a retrospective group that had already received the intervention (n = 28) as a 36-week follow-up, and a prospective intervention group (n = 22). The educational intervention had a significant positive effect on the two intervention groups' VLU HL compared to baseline. Recurrence rates appeared to be lower in the two intervention groups compared to the control group. VLU healing rates were not reported. Notably, across both studies specific HL pre-intervention could be characterised as poor, with participants scoring on average between 4.1 and 4.3 correct answers out of 13.

**Quality assessment.** We used four various critical appraisal tools recommended for each included study design (Please refer to the individual quality assessment tools–supplementary files 1–5 in S1 File) and, therefore, a summary table was not generated.

The quality of a small scale double-blind randomised controlled trial by Clarke Moloney et al [46] was rated as poor. Although the researchers appropriately reported their attempts to minimize bias, they had a small sample of 20 participants, and the probability of committing the type 1 error is high. The quality of the quasi-randomised controlled trial by Protz et al [47] was assessed as fair. The researchers were not blinded and, therefore, there is a high risk of selection bias. There was no information on any differences between the study groups that could affect the outcomes reported at baseline. The cost-effectiveness analysis of this intervention was not conducted.

The quality of the cross-sectional study [51] was rated as good. The quality of the pre- and post- educational intervention study [49] and a non-randomised controlled intervention [50] were assessed as poor due to the small sample size and a high probability of the type 1 error. Moreover, the study outcomes, including wound healing and recurrence were self-reported by the participants and were not assessed by clinicians. Furthermore, the study outcomes were not controlled for various comorbidities, which might have influenced the VLU healing and recurrence rather than improved knowledge.

## Discussion

This review was aimed to scope the literature that examined HL (both broad and disease specific) in people with VLUs, and how HL might link to self-management (particularly compression adherence and physical activity) knowledge and healing outcomes. Limited conclusions could be reached from these studies, as there were few that met the criteria overall. The included studies suggested that VLU specific HL was low, and disease-specific HL interventions may have some impact on specific HL and wound healing. Therefore, it is possible that HL does have some link with compression utilisation and wound healing, but the evidence is limited.

The included study that examined HL in VLU patients suggests that specific HL is low [51]. Overall, most studies suggested that VLU-specific HL was far below what would be needed in order to manage one's condition. There is no research using a standardized general HL tool in VLU patients, like the well validated HLS-EU-Q16 [52]. However, by focusing on specific HL on a debilitating condition that consumes a large amount of time and effort by patients [53], arguably this may be superior than a general intervention that may not add VLU specific skills in some environments. For example, in Australia, contacting one's general practitioner and following their recommended treatments may have utility for heart disease, but asking about VLUs would most likely lead to incorrect treatment advice from General Practitioners (GPs) as GP knowledge on VLU treatment is low [7, 54]. Furthermore, even if GPs did offer correct information, there is a gap between patient and GP motivations in VLU management, meaning this gap likely also affects attending to specific HL advice [55]. Therefore, assessing VLU specific HL may be a superior assessment.

There were no identified research studies that examined the correlation between HL and the predictors of adherence to physical activity, and VLU wound healing. Only one correlational study showed VLU specific HL was associated with greater self-reported compression rates [51]. Intervention-based research suggest there may be some utility in improving HL to improve patient activities in compression and therefore, wound healing. Most notably, the two American studies by Gonzales [49, 50] provided the strongest link, finding that improving VLU specific HL improved knowledge and the likelihood of recurrence. While VLU healing was not affected, this was likely due to the smaller sample size (i.e., smaller or medium effect sizes undetectable in smaller samples) or the delayed effect of HL improvement on VLU improvement.

The lack of research on physical activity and HL in VLUs was surprising, but may be due to the significant controversy in whether or not physical activity, in addition to compression, has any effect [17]. Although there is strong theoretical evidence for physical activity in improving VLU healing [56], the link may be weaker or more distal than other factors [17], and thus fewer studies may seek to examine this factor relative to more direct links.

Overall, the research into HL and VLUs is too limited to make any assertions on the state of HL in VLU patients, and how HL may relate to compression, healing, or physical activity. Recent health belief research on patients with other types of ulcers with similar risk profiles indicated that prior assumptions and knowledge were integral in adherence to health directives [57]. The little evidence that exists on HL and VLUs seems to have similar implications.

## Limitations

The main limitation of this scoping review is that all included studies have assessed the participants' knowledge rather than health literacy, which limited the relevance of the included studies to our research questions. In fact, there was overlap between VLU knowledge and VLU HL skills, which the authors had to detangle through consensus. Three research questions we formulated in this review did not match well with the research questions of the included studies. There is a considerable dearth of high quality, large observational studies that examine HL and VLU healing over time, and the factors involved. Of all the studies included in this review, only two examined VLU-specific HL and healing, although neither reported these findings in sufficient detail, and both had small sample sizes. Both were also written by the same author in one region of the US. The reported findings have limited generalisability. The research questions we formulated in this review, and the research questions in each study included in this review, largely could not be answered with the findings the authors provided. Furthermore, there are no unified HL scales used across studies on VLUs (either general or specific),

meaning it is difficult to compare across studies. In fact, there was overlap between VLU knowledge and VLU HL skills, which the authors had to detangle through consensus.

Most of the included studies were experimental, meaning many required smaller sample sizes to assess their research question compared to ours (as effect sizes are larger for within subject experimental studies than observational ones). By contrast, in order to optimally assess our research question, we would need to assess a wide variety of studies with somewhat similar conceptualisations on VLU specific HL, following many patients over a long period of time. A large, prospective study using well-validated tools of HL and direct observation of compression adherence, physical activity, and VLU healing could assist in answering our research questions.

Our terminology in the research question required far more judgement of what constituted HL versus health knowledge than initially anticipated. While our protocol suggested we would include health knowledge to some degree, it became clear (through our assessments) that HL and health knowledge were often tangled in these papers. Therefore, authors had to make a judgement on HL versus health knowledge, a concept not otherwise included in the protocol. In the protocol, we stated that we will use CASP to assess the quality of the included studies. However, because of the various study designs of the included studies, we also used other critical appraisal tools recommended for quality assessment of these designs. While these deviations likely did not significantly affect the results, nor the process, it is important to clarify our decisions to ensure clarity of our findings.

Finally, there is a fundamental assumption embedded in the theoretical approach of our research question: an implied mediation that none of the studies cited could directly answer. That is, a key part of the study rationale posits that HL leads to improved self-management through compression and physical activity, which then leads to faster VLU healing. However, this link is not assessed in this order in any of the studies, as it would require a longitudinal study with strong causal mechanisms being demonstrated between each of these factors over time. Furthermore, this study would have to also assess other possible causal factors that may correlate with HL which leads to higher compression rates. For example, patients from higher socioeconomic groups with access to higher quality healthcare may gain specific and general HL through higher quality care, or through the reverse [58]. There is a potential issue where those who have higher socioeconomic status and who have higher levels of HL may also be better able to find and process patient education materials, meaning HL and patient education may be conflated in this paper. Future research should seek to clarify any potential links between HL, socioeconomic status, adherence to management and treatment regimens, and healing by incorporating a qualitative component to a large-scale study. For example, using an established consumer wound group made up of families and caregivers would assist greatly in understanding the links between HL and VLU specific outcomes.

## Conclusions

The results of this review suggest that there is a lack of high-quality research on the links between HL and VLU patients' compression utilisation and VLU healing. Limited research suggests that there is a deficit of HL in VLU patients. While the effects of HL have received more attention in recent years in other conditions, such as diabetic foot ulcers, surprisingly little research has been conducted on the role of HL in VLU management. We, therefore, highlight the need for a quality prospective study that employ validated HL tools to assess HL in patients with VLUs.

## Supporting information

**S1 Checklist. Preferred Reporting Items for Systematic reviews and Meta-Analyses extension for Scoping Reviews (PRISMA-ScR) checklist- for health literacy and VLU healing.**
(DOCX)

**S1 Appendix. Search strings for MEDLINE via Ovid.**
(DOCX)

**S1 File.**
(ZIP)

## Acknowledgments

Monash Library
    Catelyn Richards (research assistant)

## Author Contributions

**Conceptualization:** Carolina Weller.

**Data curation:** Ayoub Bouguettaya, Georgina Gethin, Jane Sixsmith.

**Formal analysis:** Ayoub Bouguettaya, Georgina Gethin, Sebastian Probst, Victoria Team.

**Investigation:** Ayoub Bouguettaya, Carolina Weller.

**Methodology:** Ayoub Bouguettaya, Sebastian Probst.

**Project administration:** Georgina Gethin, Sebastian Probst, Jane Sixsmith, Carolina Weller.

**Supervision:** Georgina Gethin, Victoria Team, Carolina Weller.

**Validation:** Ayoub Bouguettaya, Jane Sixsmith.

**Visualization:** Ayoub Bouguettaya.

**Writing – original draft:** Ayoub Bouguettaya.

**Writing – review & editing:** Ayoub Bouguettaya, Georgina Gethin, Sebastian Probst, Jane Sixsmith, Victoria Team, Carolina Weller.

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
