## [Decision Letter · Decision Letter 0]

10 Mar 2022

PONE-D-21-40133How health literacy relates to venous leg ulcer healing: A scoping review.PLOS ONE

Dear Dr. Bouguettaya,

Thank you for submitting your manuscript to PLOS ONE. After careful consideration, we feel that it has merit but does not fully meet PLOS ONE’s publication criteria as it currently stands. Therefore, we invite you to submit a revised version of the manuscript that addresses the points raised during the review process.

We look forward to receiving your revised manuscript.

Kind regards,

Professor dr. Dimitri Beeckman, Ph.D.

Journal Requirements:

- https://bmjopen.bmj.com/content/bmjopen/11/5/e044604.full.pdf?with-ds=yes

In your revision ensure you cite all your sources (including your own works), and quote or rephrase any duplicated text outside the methods section. Further consideration is dependent on these concerns being addressed.

“The senior author (CW) was supported by the grant from the Australian NHMRC (NUMBER APP1069329). Funders had no role in this manuscript.”

“The corresponding author (CW) was supported by the grant from the Australian NHMRC (NUMBER APP1069329).

https://www.nhmrc.gov.au/

he funders had no role in study design, data collection and analysis, decision to publish, or preparation of the manuscript.”

Additional Editor Comments:

Thank you very much for submitting your manuscript. We have now received the reports from two reviewers and both have positive and negative comments. Some comments are also somewhat contradictory. I agree with reviewer #1 that the concepts (health literacy and knowledge) should be more clearly defined. They are the basis for this paper. I think the comment about the different dimensions is important. The other comments, of course, fit seamlessly into this feedback and deserve the necessary attention/focus. I encourage the authors to revise their paper and submit an amended version.

Reviewers' comments:

Reviewer's Responses to Questions

**Comments to the Author**

1. Is the manuscript technically sound, and do the data support the conclusions?

Reviewer #1: No

Reviewer #2: Yes

2. Has the statistical analysis been performed appropriately and rigorously? 

Reviewer #1: N/A

Reviewer #2: N/A

3. Have the authors made all data underlying the findings in their manuscript fully available?

Reviewer #1: Yes

Reviewer #2: Yes

4. Is the manuscript presented in an intelligible fashion and written in standard English?

Reviewer #1: Yes

Reviewer #2: Yes

5. Review Comments to the Author

Reviewer #1: Thank you for the opportunity to read this manuscript. I have read it with great interest because I think this is an important topic to investigate.

I have read the manuscript and it puzzles me because it features two concepts ( knowledge and health literacy). I think it is important to be distinct with these concepts especially as the purpose of this scoping review was to investigate HL and VLU. The two concepts are not clearly described at this stage. Consider revising aim and result and discussion.

Introduction

Describes VLU very well. But the section about HL needs to be elaborated. The authors write about general and specific HL. This way of describing HL is new to me. To my knowledge , HL levels refer to three dimension functional, communicative or critical ( Nutbeam, 2009). I wonder if the authors mean generic or disease specific instruments? However, this categorisations or levels need to explained further.

I lack information about the different dimensions in HL: functional, communicative and critical HL or functional/basic, communicative/interactive and critical HL depending on different definitions. Also, a description of what these dimensions entails.

Further, FHL, C& C HL scale by Ishikawa was first developed and tested in a sample of patients with diabetes. Ishikawas instruments have been translated to different languages and have been tested in different populations (eg. to mention some, office workers, patients with Parkinson, patients with obesity). The original FHL and CC & C HL by Ishikawa was valid and reliable in a group with diabetes. But the questions were not explicitly linked to diabetes ( example of one question in FHL scale: Do you think that it is difficult to read health information because the text is difficult to see (even if you have glasses or contact lenses)? Therefore, I wonder if the “etiquette” that it is a disease specific instrument is perhaps not fully correct.

I lack information on how to measure HL, there are a variety of instruments available.

Suggestion of references to read more about the concept of HL

Nutbeam, D. (2000). Health literacy as a public health goal: a challenge for contemporary health education and communication strategies into the 21st century. Health promotion international, 15(3), 259-267.

Nutbeam, D. (2009). Defining and measuring health literacy: what can we learn from literacy studies?. International journal of public health, 54(5), 303-305.

Nutbeam, D., & Muscat, D. M. (2020). Advancing health literacy interventions. In Health Literacy in Clinical Practice and Public Health (pp. 115-127). IOS Press.

Sørensen, K., Van den Broucke, S., Fullam, J., Doyle, G., Pelikan, J., Slonska, Z., & Brand, H. (2012). Health literacy and public health: a systematic review and integration of definitions and models. BMC public health, 12(1), 1-13.

Line 127 In the rationale, “… specific HL in VLU is poor…(32). This study investigates knowledge and educational needs, but the study does not use a valid and reliable HL instrument to evaluate HL. In fact, the concept is not at all mentioned in this article. Therefore, I do not think this article should be referred to in the rationale.

Method

When reading the description of the five included studies, no one seem to have HL as an outcome measure. Non seem to have valid and reliable instruments to measure HL. Instead included articles have outcome measures of knowledge. The authors need to explain the choice of eligible articles, and if the included articles in fact measures HL or if the measure knowledge after an educational intervention.

The alignment between aim, research questions and result is not clear to me. Also, the categorisation between general and specific HL puzzles me. I expect to read about sufficient, problematic and inadequate levels of HL and this is not mentioned at all in the result.

The result section would be easier to read if there were subheadings, the same as the three research questions.

Discussion may need to be revised after clarification of concepts that are used.

Reviewer #2: ABSTRACT:

The abstract is clear and reflect the findings and conclusions of the review.

BACKGROUND, OBJECTIVES, AND METHODS:

The background is clear and concise. It establishes the context of the review. The authors might want to consider an update of some of the references – for example, there is new data in estimating venous leg ulcer prevalence/complex wounds prevalence.

The primary research question is clear. The secondary research question addresses the relationship between two variables – I would suggest explaining what kind of relationship.

The methods section is well written. Review is reported in the line with PRISMA ScR. Protocol was published in advance (Weller et al., 2021). The authors have built this scoping review using Levac et al. and have described proposed methods in a Protocol (Weller et al., 2021), however there is a mismatch between the protocol and the manuscript. It relates to the critical appraisal (Stage 5 of Levac et al.). I suggest that authors discuss these differences and explain how/if they have carried out critical appraisal using CASP (as written in the Protocol). This might be important, as the authors have excluded 1 record based on unclear method and 14 records based on unclear/lack of details.

RESULTS:

The studies included were clearly presented. Relevant data was reported in the text and the tables/appendix. There is a mismatch between the Protocol (Weller et al., 2021) and the manuscript (CASP evidence profile table is missing). I would suggest that authors discuss differences.

p10/l210,211 - consider adding details/result of statistic test.

p11/l236 compression adherence/self management?

Tables:

Table 1: Included studies

p20/results column - if this is p value - please add to the result.

p.28/results column - please omit the question (?How much data to pull from this paper) and details of results.

DISCUSSION:

The discussion is well written. The authors have appropriately interpreted the results of the review. They discuss the completeness and applicability evidence.

Limitations:

p12/l273 - It is not clear how/what significant limitation impacted the scoping review - perhaps it is only a language issue (the language is unclear, making it difficult to follow). Scoping review has identified the lack of evidence, which is fine and it emphasised what needs to be done in the future. You could not answer at your research questions in full due to i. lack of evidence and ii. certainty of evidence. Lack of evidence should not be confused with the certainty of evidence.

CONCLUSION:

The implications for practice and research are clear and concise, which helps the reader to receive a clear message. The authors are cautious in their implications; however, this matches the certainty of the evidence.

I would suggest considering rephrasing the first sentence - there is a lack of high quality research - there is a general lack of research.

Typographical errors:

p4/l73 - consistent-the, dash instead of hyphen

p5/l91 - delete last bracket

6. PLOS authors have the option to publish the peer review history of their article (what does this mean?). If published, this will include your full peer review and any attached files.

Reviewer #1: No

Reviewer #2: No

---

## [Author Response · Author response to Decision Letter 0]

9 May 2022

Reviewer 1’s comments:

Reviewer #1: Thank you for the opportunity to read this manuscript. I have read it with great interest because I think this is an important topic to investigate.

I have read the manuscript and it puzzles me because it features two concepts (knowledge and health literacy). I think it is important to be distinct with these concepts especially as the purpose of this scoping review was to investigate HL and VLU. The two concepts are not clearly described at this stage. Consider revising aim and result and discussion.

Thank you for these suggestions and feedback, it was helpful. We have amended the manuscript in line with these comments. 

Introduction

Describes VLU very well. But the section about HL needs to be elaborated. The authors write about general and specific HL. This way of describing HL is new to me. To my knowledge, HL levels refer to three dimension functional, communicative or critical (Nutbeam, 2009). I wonder if the authors mean generic or disease specific instruments? However, this categorisations or levels need to explained further.

I lack information about the different dimensions in HL: functional, communicative and critical HL or functional/basic, communicative/interactive and critical HL depending on different definitions. Also, a description of what these dimensions entails.

We admit that we did consider splitting the research against these ideas, only not doing so for word count and for the clarity of the paper. This research’s explanation of health literacy in this way was in fact inspired by that same 2009 paper, in which the authors argued that health literacy needs to be understood as “content-specific literacy in a health context.” Nutbeam’s 2000 paper describing these three concepts of function, communication, and critical are described in this paper as “general literacy”, with those domains listed within them. This split is also partially inspired by Bailey et al 2014- on diabetes scales. Bailey splits the scales used in research as general vs specific, and we sought to follow the same pattern. 

However, to address this, we have added a few lines explaining general HL against Nutbeam’s 2000, and 2009 papers and our explanation, on lines 100-107.

Bailey SC, Brega AG, Crutchfield TM, Elasy T, Herr H, Kaphingst K, Karter AJ, Moreland-Russell S, Osborn CY, Pignone M, Rothman R. Update on health literacy and diabetes. The Diabetes Educator. 2014 Sep;40(5):581-604.

Further, FHL, C& C HL scale by Ishikawa was first developed and tested in a sample of patients with diabetes. Ishikawas instruments have been translated to different languages and have been tested in different populations (eg. to mention some, office workers, patients with Parkinson, patients with obesity). The original FHL and CC & C HL by Ishikawa was valid and reliable in a group with diabetes. But the questions were not explicitly linked to diabetes (example of one question in FHL scale: Do you think that it is difficult to read health information because the text is difficult to see (even if you have glasses or contact lenses)? Therefore, I wonder if the “etiquette” that it is a disease specific instrument is perhaps not fully correct.

This was an error; the citation should have been Nath et al. and Yamashita 2011. We have corrected this citation. 

I lack information on how to measure HL, there are a variety of instruments available.

Suggestion of references to read more about the concept of HL

Nutbeam, D. (2000). Health literacy as a public health goal: a challenge for contemporary health education and communication strategies into the 21st century. Health promotion international, 15(3), 259-267.

Nutbeam, D. (2009). Defining and measuring health literacy: what can we learn from literacy studies?. International journal of public health, 54(5), 303-305.

Nutbeam, D., & Muscat, D. M. (2020). Advancing health literacy interventions. In Health Literacy in Clinical Practice and Public Health (pp. 115-127). IOS Press.

Sørensen, K., Van den Broucke, S., Fullam, J., Doyle, G., Pelikan, J., Slonska, Z., & Brand, H. (2012). Health literacy and public health: a systematic review and integration of definitions and models. BMC public health, 12(1), 1-13.

We’ve included the requested information and added the suggested citations in the relevant sections. 

Line 127 In the rationale, “… specific HL in VLU is poor…(32). This study investigates knowledge and educational needs, but the study does not use a valid and reliable HL instrument to evaluate HL. In fact, the concept is not at all mentioned in this article. Therefore, I do not think this article should be referred to in the rationale.

We have replaced this HL with Health knowledge which might address this. 

Method

When reading the description of the five included studies, no one seem to have HL as an outcome measure. Non seem to have valid and reliable instruments to measure HL. Instead included articles have outcome measures of knowledge. The authors need to explain the choice of eligible articles, and if the included articles in fact measures HL or if the measure knowledge after an educational intervention. The alignment between aim, research questions and result is not clear to me. Also, the categorisation between general and specific HL puzzles me. I expect to read about sufficient, problematic and inadequate levels of HL and this is not mentioned at all in the result.

We have modified the aim and questions somewhat to incorporate knowledge, and added a bit of discussion on the issues of knowledge vs HL (lines 88-92). The problem is that the way knowledge is explained in these studies. To give an example, questions on self-care and capacity, and self-efficacy would fall under the ability to navigate one’s environment and use information, which we interpreted as HL. So, we had to make a judgement in many cases as to what the authors of each of the included studies were actually measuring- consensus reached through conversation between the authors.

The result section would be easier to read if there were subheadings, the same as the three research questions.

We have added a bit on the results by splitting into subheadings. 

Discussion may need to be revised after clarification of concepts that are used.

We have modified our discussion, reflecting more on our research questions and discussing the limitations of this review. 

Reviewer 2’s comments: 

ABSTRACT:

The abstract is clear and reflect the findings and conclusions of the review.

BACKGROUND, OBJECTIVES, AND METHODS:

The background is clear and concise. It establishes the context of the review. The authors might want to consider an update of some of the references – for example, there is new data in estimating venous leg ulcer prevalence/complex wounds prevalence.

Thank you. We have added the most recent studies, estimating global VLU prevalence and describing epidemiological profile of the VLU patients, as well as other studies estimating venous leg ulcer prevalence/complex wounds prevalence.

The primary research question is clear. The secondary research question addresses the relationship between two variables – I would suggest explaining what kind of relationship.

We have added the statement (correlational or experimental).

The methods section is well written. Review is reported in the line with PRISMA ScR. Protocol was published in advance (Weller et al., 2021). The authors have built this scoping review using Levac et al. and have described proposed methods in a Protocol (Weller et al., 2021), however there is a mismatch between the protocol and the manuscript. It relates to the critical appraisal (Stage 5 of Levac et al.). I suggest that authors discuss these differences and explain how/if they have carried out critical appraisal using CASP (as written in the Protocol). This might be important, as the authors have excluded 1 record based on unclear method and 14 records based on unclear/lack of details.

Thank you for your comment. We have clarified that we have excluded 18 studies when assessing full text records. We have included the quality assessment sub-section, where we discussed how we have carried out critical appraisal of the included studies (Pages 12-13).

RESULTS:

The studies included were clearly presented. Relevant data was reported in the text and the tables/appendix. There is a mismatch between the Protocol (Weller et al., 2021) and the manuscript (CASP evidence profile table is missing). I would suggest that authors discuss differences.

We have added a sub-section on the quality assessment. We used CASP for RCTs as we discussed in the protocol, and other relevant critical appraisal tools. The differences between the protocol and the study were discussed.

In the protocol, we stated that we will use CASP to assess the quality of the included studies. However, because of the various study designs of the included studies, we also used other critical appraisal tools recommended for quality assessment of these designs.

DISCUSSION:

The discussion is well written. The authors have appropriately interpreted the results of the review. They discuss the completeness and applicability evidence.

Limitations:

p12/l273 - It is not clear how/what significant limitation impacted the scoping review - perhaps it is only a language issue (the language is unclear, making it difficult to follow). Scoping review has identified the lack of evidence, which is fine and it emphasised what needs to be done in the future. You could not answer at your research questions in full due to i. lack of evidence and ii. certainty of evidence. Lack of evidence should not be confused with the certainty of evidence.

Thank you. We have restructured our Limitations section and clarified the main limitations of our review. These include: 1) subjective evaluation of ‘knowledge’ and ‘health literacy concepts’; 2) ability to address the research question; and 3) deviations from the protocol.

CONCLUSION:

The implications for practice and research are clear and concise, which helps the reader to receive a clear message. The authors are cautious in their implications; however, this matches the certainty of the evidence.

I would suggest considering rephrasing the first sentence - there is a lack of high-quality research - there is a general lack of research.

Thank you for this comment. We’ve removed this. 

Typographical errors:

p4/l73 - consistent-the, dash instead of hyphen

p5/l91 - delete last bracket

Corrected

p10/l210,211 - consider adding details/result of statistic test.

p11/l236 compression adherence/self-management?

Added

Tables:

Table 1: Included studies

p20/results column - if this is p value - please add to the result.

p.28/results column - please omit the question (?How much data to pull from this paper) and details of results.

Completed.

---

## [Decision Letter · Decision Letter 1]

6 Jun 2022

PONE-D-21-40133R1How health literacy relates to venous leg ulcer healing: A scoping review.PLOS ONE

Dear Prof. Weller,

Dear Carolina 

Thank you for submitting your manuscript to PLOS ONE. All comments have been considered and the manuscript is almost ready for acceptance. There is still one comment from the reviewer (place of quality assessment of reporting) that I think is important and requires the necessary attention. In addition, there are some typographical comments. I am happy to expedite the review when the manuscript is resubmitted and make a final decision ASAP. Therefore, I invite you to submit a revised version of the manuscript that addresses the minor points raised.

We look forward to receiving your revised manuscript.

Kind regards,

Professor Dimitri Beeckman, PhD

Journal Requirements:

Additional Editor Comments:

After a thorough review by the reviewer and by me, I am pleased to inform you that your manuscript is almost ready for acceptance. There is still one comment made by the reviewer (place of reporting quality assessment) that I think is important and requires the necessary attention. In addition, there are some typographical comments. May I ask you to revise the manuscript and upload a new version. I will be happy to review it myself before accepting the manuscript for publication. It is a very interesting manuscript and I would like to thank you for submitting it to PLOS ONE.

Reviewers' comments:

Reviewer #2: All comments have been addressed. However, I suggest that authors consider dividing description of procedures of critical appraisal (add to Methods section) from reporting the quality of evidence (which should stay in Results section).

Typographical errors:

line 92, remove one dot (hold. .)

line 103, add dot after reference (20).

line 106, add dot after reference (17, 20).

line 124, add dot after reference (26).

---

## [Author Response · Author response to Decision Letter 1]

3 Dec 2022

Dear Professor Dimitri Beeckman,

Thank you for your comments and assistance on this manuscript. 

Here, we outline our responses to the letter.

Reviewer 2:

Reviewer #2: All comments have been addressed. However, I suggest that authors consider dividing description of procedures of critical appraisal (add to Methods section) from reporting the quality of evidence (which should stay in Results section).

Thank you for this comment. We have addressed this by moving those sections into the relevant place, into the methods. 

Typographical errors:

line 92, remove one dot (hold. .)

line 103, add dot after reference (20).

line 106, add dot after reference (17, 20).

line 124, add dot after reference (26).

We have addressed all these by making these exact changes.

---

## [Editor Report · Decision Letter 2]

7 Dec 2022

How health literacy relates to venous leg ulcer healing: A scoping review.

PONE-D-21-40133R2

Dear Dr. Bouguettaya,

We’re pleased to inform you that your manuscript has been judged scientifically suitable for publication and will be formally accepted for publication once it meets all outstanding technical requirements.

Kind regards,

Professor Dimitri Beeckman, Ph.D.

Additional Editor Comments (optional):

Thank you very much - this manuscript is now ready for publication. Congratulations.
---

## [Editor Report · Acceptance letter]

6 Jan 2023

PONE-D-21-40133R2 

How health literacy relates to venous leg ulcer healing: A scoping review. 

Dear Dr. Bouguettaya:

I'm pleased to inform you that your manuscript has been deemed suitable for publication in PLOS ONE. Congratulations! Your manuscript is now with our production department. 

Kind regards, 

on behalf of

Professor Dimitri Beeckman 

Academic Editor

PLOS ONE